# Psychological impact of the COVID-19 epidemic among healthcare workers in paediatric intensive care units in China

**Yue Zhang, Dan-Dan Pi, Cheng-Jun Liu, Jing Li, Feng Xu\***

Department of Pediatric Intensive Care Unit, Ministry of Education Key Laboratory of Child Development and Disorders, National Clinical Research Center for Child Health and Disorders, China International Science and Technology Cooperation Base of Child Development and Critical Disorders, Children's Hospital of Chongqing Medical University, Chongqing, P.R China

\* xufeng9899@163.com

## Abstract

To perform a mental health evaluation and an early psychological intervention for healthcare workers (HCWs) during the coronavirus disease 2019 (COVID-19) epidemic, an online survey was conducted among 3055 HCWs in the paediatric intensive care units (PICUs) of 62 hospitals in China on March 26, 2020, by the Neurology and Sedation Professional Group, Emergency Department, Paediatrics Branch, Chinese Medical Association. The questionnaire was divided into three parts, including general information, the Impact of Event Scale-Revised (IES-R), and the Depression Anxiety Stress Scale-21 (DASS-21). The results show that a total of 970 HCWs (45.99%) were considered to meet the clinical cut-off scores for posttraumatic stress (PTS), and the proportions of participants with mild to extremely severe symptoms of depression, anxiety and stress were 39.69%, 36.46% and 17.12%, respectively. There was no significant difference in the psychological impact among HCWs of different genders. Married HCWs were 1.48 times more likely to have PTS than unmarried HCWs (95% CI: 1.20–1.82, p <0.001). Compared with junior professional title participants, the PTS-positive rate of HCWs with intermediate professional titles was 1.91 times higher (90% CI: 1.35–2.70, p<0.01). Those who had been in contact with confirmed COVID-19 cases were 1.40 times (95% CI: 1.02–1.92, p <0.05) more likely to have PTS than those who did not have contact with COVID-19 cases or did not know the relevant conditions. For depression, the proportion of HCWs with intermediate professional titles was significantly higher, at 1.65 times (90% CI: 1.17–2.33, p <0.01) that of those with junior professional titles. The depression of HCWs at work during the epidemic was 1.56 times that of HCWs on vacation (95% CI: 1.03–2.37, p <0.05), and their anxiety was 1.70 times greater (95% CI: 1.10–2.63, p <0.05). Participants who had been in contact with confirmed COVID-19 cases had more pronounced anxiety, 1.40 times that of those who did not have contact with COVID-19 cases or did not know the relevant conditions (95% CI: 1.02–1.92, p <0.05). There was no significant correlation between the variables and the positive results of stress symptoms. In total, 45.99%, 39.69%, 36.46% and 17.12% of PICU HCWs were affected by PTS, depression, anxiety and stress, respectively, to varying degree. Married status, intermediate professional titles and exposure history were independent risk factors for PTS. Intermediate

**Data Availability Statement:** All relevant data are within the paper and its Supporting information files.

**Funding:** The author(s) received no specific funding for this work.

**Competing interests:** The authors have declared that no competing interests exist.

professional titles and going to work during the epidemic were independent risk factors for depression, and going to work and exposure history during the epidemic were independent risk factors for anxiety. In the face of public health emergencies, HCWs not only specialize in paediatric intensive care but also, as a high-risk group, must actively take preventive measures and use mitigation strategies.

## Introduction

Coronavirus disease 2019 (COVID-19), or acute respiratory disease caused by severe acute respiratory syndrome coronavirus 2 (SARS-CoV-2), was first discovered in Wuhan, China, and referred to as "new coronary pneumonia" [1]. It was named by the World Health Organization on February 11, 2020 [2], and included in category B infectious diseases by the National Centre for Disease Control and Prevention, and prevention and control measures for category A infectious diseases were adopted [3]. COVID-19 is mainly spread through the respiratory tract. Unlike severe acute respiratory syndrome coronavirus (SARS-CoV) and Middle East respiratory syndrome coronavirus (MERS-CoV), SARS-CoV-2 is more infectious [4]. As of March 26, 2020, SARS-CoV-2 had caused more than 80,000 people to be infected and more than 3,000 people to die in China. At that time, the COVID-19 outbreak in China was basically under control, but countries such as Europe and the United States were experiencing outbreaks [5–10] (S1 and S2 Figs).

To reduce the flow of people and control the spread of new coronary pneumonia, on January 22, the Central Committee of the Communist Party of China decisively required Hubei Province to implement comprehensive and strict control over the outflow of people [4]. Healthcare workers (HCWs) from all over the country successively travelled to Wuhan to provide support [4], and it was inevitable that some would contact with suspected or confirmed cases of COVID-19. Most HCWs stayed in the hospital, but it was easy for the general public to come into contact with patients with respiratory symptoms such as fever and cough that could not immediately be ruled out as COVID-19. According to reports, special groups such as frontline healthcare workers, the elderly, children, college students, the LGBTQ+ community, homeless and economically vulnerable individuals, rural communities, foreigners and psychiatric patients were more vulnerable to mental health effects [11–14]. Psychological distress of general public might have been directly caused by restrictive strategies and reduced social mobility [15–18], while HCWs' distress was often caused by fear of being infected and infecting others, higher workload, significant pressure, pain of losing patients and colleagues, the still-unpredictable nature of the virus, inadequate testing, limited treatment options and disruption of regular routine, and shortages in personal protective equipment and other medical supplies [17, 19, 20]. With past public health emergencies, such as SARS in 2002 and MERS in 2012, many HCWs suffered emotional distress and mental trauma and have long-term effects [21–24]. HCWs as a high-risk group, we inferred that COVID-19 is also likely to produce varying degrees of negative emotional symptoms among this population. Coupled with the fact that COVID-19 is more likely to produce severe cases than previous pandemics [25], this epidemic presents many challenges for HCWs in the intensive care unit.

According to the data currently available, the infection and prevalence of COVID-19 in children is not very clear, and some do not believe that COVID-19 can infect children, or if it can, that its severity rate in children is extremely low [26]. This uncertainty also presents challenges for paediatric intensive care units (PICU) HCWs. At present, research on the

psychological effects of PICU HCWs is very limited, and there are not enough sample data to report the psychological effects of the outbreak of COVID-19. To study the psychological impact of the COVID-19 outbreak on HCWs and analyse their independent risk factors, the Emergency Department of the Paediatrics Branch of the Chinese Medical Association investigated the mental health status of HCWs in PICUs across the country immediately after the COVID-19 epidemic was basically controlled in China to provide a reference for countries to conduct psychological interventions for HCWs as early as possible.

## Materials and methods

This study is a multicentre, cross-sectional online survey. Expedited ethics approval was obtained from the Institutional Review Board, Children's Hospital of Chongqing Medical University (CHCQMU-IRB-2020-304), which conformed to the principles embodied in the Declaration of Helsinki. The online questionnaire was sent to 62 hospitals in 31 provinces (municipalities or autonomous regions) of China on March 26, 2020. The questionnaires were distributed to a total of 3055 HCWs in these 62 hospitals, and a total of 2116 questionnaires were collected on April 15, 2020. Seven questionnaires were excluded due to improper completion, leaving a total of 2109 questionnaires. Since the questionnaire was completed voluntarily, the response rate was not calculated. All participants voluntarily responded to the survey anonymously and provided informed consent online before the survey.

The questionnaire is divided into three parts. ① General information: age, gender, marital status, residence, specialty, PICU experience, employment title, education attainment, and questions, including "Are you still working during the epidemic?", "Do you have contact with confirmed COVID-19 cases?", and "Are you sure the hospital (or PICU) has confirmed cases or the isolation ward has suspected cases?". ② The Impact of Event Scale-Revised (IES-R) [27, 28], including the intrusion subscale (items 1, 2, 3, 6, 9, 14, 16 and 20), avoidance subscale (items 5, 7, 8, 11, 12, 13, 17 and 22) and hyperarousal subscale (items 4, 10, 15, 18, 19 and 21). The scale uses a 5-level scoring method, with a defined score of <24 as no posttraumatic stress (PTS), 24–32 as mild PTS, 33–36 as moderate PTS, and 37–88 as severe PTS [29, 30]. ③ The Depression, Anxiety and Stress Scale-21 (DASS-21) [31] includes the depression subscale (items 3, 5, 10, 13, 16, 17, and 21), anxiety subscale (items 2, 4, 7, 9, 15, 19, and 20) and stress subscale (items 1, 6, 8, 11, 12, 14, and 18). The subscale scores can be allocated to one of 5 levels of severity: for depression, normal (0–4), mild (5–6), moderate (7–10), severe (11–13), and extremely severe (14–21); for anxiety, normal (0–3), mild (4–5), moderate (6–7), severe (8–9), and extremely severe (10–21); and for stress, normal (0–7), mild (8–9), moderate (10–12), severe (13–16), and extremely severe (17–21). The Chinese versions of the IES-R and DASS-21 have been shown to have good reliability and validity [32–37].

In this study, statistical analysis was performed using SPSS Statistic 25.0 (IBM SPSS Statistics, New York, United States). The count data are expressed as percentages, and the measurement data are expressed as averages and standard deviations. T-tests, F-tests, chi-square tests, and binary logistic regression were used to analyse the data. Statistical significance of all the two-tailed tests was set at $p < 0.05$.

## Results

A total of 2109 HCWs completed the survey, of whom 85.02% (1793/2109) were female and 14.98% (316/2109) were male. Participants ranged in age from 20 to 60 years old, with an average age of 32.42 (SD = 6.66). A total of 739 HCWs (35.04%) were doctors, and 1370 HCWs (64.96%) were nurses. During the epidemic, more than 90% (1992/2109) of HCWs were still at work, of whom 20.8% (416/1992) remained on the front lines; 216 participants went to Wuhan

**Table 1. Socio-demographic characteristics of participants (n = 2109).**

| Variables | | N | % |
|---|---|---|---|
| Age(Years) | 20–29 | 750 | 35.56 |
| | 30–49 | 1309 | 62.07 |
| | 50–60 | 50 | 2.37 |
| Gender | Male | 316 | 14.98 |
| | Female | 1793 | 85.02 |
| Marital status | Unmarried | 653 | 30.96 |
| | Married | 1456 | 69.04 |
| Residence | Others | 2008 | 95.21 |
| | Wuhan | 101 | 4.79 |
| Specialty | Doctor | 739 | 35.04 |
| | Nurse | 1370 | 64.96 |
| PICU experience(Years) | <1 | 253 | 12 |
| | 1–10 | 1474 | 69.89 |
| | >10 | 382 | 18.11 |
| Employment title | Junior | 1288 | 61.07 |
| | Intermediate | 614 | 29.11 |
| | Senior | 207 | 9.82 |
| Education attainment | Doctorate | 57 | 2.7 |
| | Masters | 428 | 20.29 |
| | Bachelors | 1624 | 77 |
| Still working during the epidemic | No | 117 | 5.55 |
| Workplace | Yes | 1992 | 94.45 |
| | General ward or clinic | 1576 | 79.1 |
| | Isolation ward or fever clinic | 200 | 10.0 |
| | Wuhan or designated hospital | 216 | 10.8 |
| Contact with COVID-19 cases | No or not sure | 1869 | 88.62 |
| | Yes | 240 | 11.38 |
| Confirmed cases in the hospital | No or not sure | 1413 | 67 |
| | Yes | 696 | 33 |
| Confirmed cases in PICU | No or not sure | 1933 | 91.65 |
| | Yes | 176 | 8.35 |
| Suspected cases in Isolation ward | No or not sure | 671 | 31.82 |
| | Yes | 1438 | 68.18 |

or designated hospitals, and 200 participants went to isolation wards or fever clinics. The remaining baseline information is shown in Table 1.

The questionnaire contains two psychological scales: the IES-R, which is used to reflect the symptoms of PTS, and the DASS-21, whose three subscales are used to evaluate depression, anxiety and stress. A total of 970(45.99%), 837(39.69%), 769(36.46%) and 361(17.12%) participants had varying degrees of PTS and felt depression, anxiety, and stress, respectively. The severity of these conditions is shown in S1 Table.

Comparing the baseline data between groups, as shown in Fig 1 and S2 Table, there were no significant differences in psychological distress among HCWs of different genders or educational backgrounds. HCWs who were married or had interacted with suspected COVID-19 cases in the isolation ward had more PTS. In addition to having more PTS, participants who lived in Wuhan or had been exposed to COVID-19 also showed more anxiety. Doctors had more depression and stress symptoms than nurses. During the epidemic, there was no

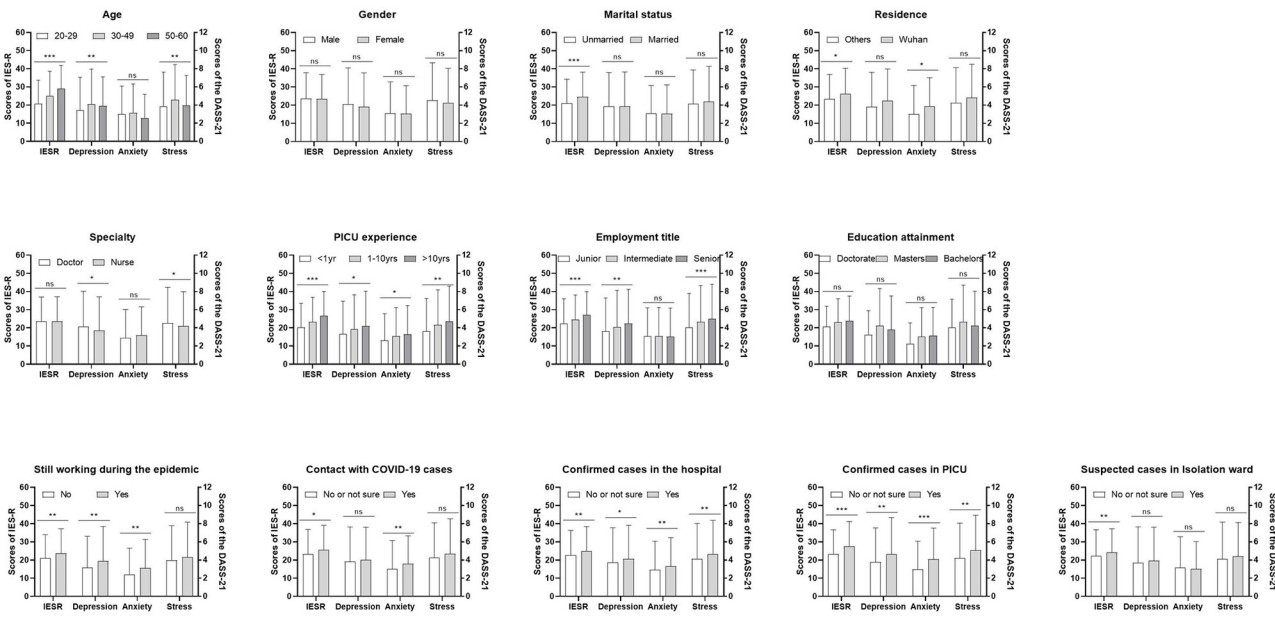

**Fig 1. Relationship between baseline characteristics and psychological changes.**

significant difference in the stress of HCWs at work compared with those on vacation, but those at work scored higher on the rest of the scale. In the IES-R and DASS-21 depression sub-scales, the scores of 30- to 49-year-old HCWs were higher than those of younger HCWs, while the scores of HCWs with intermediate professional titles were significantly higher than those of HCWs with junior professional titles. HCWs with confirmed COVID-19 cases in their hospital or PICU scored higher on each scale, while those working in the PICU for less than 1 year scored significantly lower than those working in the PICU for more than 10 years.

Fig 2 reveals that some variables are statistically associated with HCWs' PTS. Married HCWs were 1.48 times more likely to have PTS than unmarried HCWs (95% Cl: 1.20–1.82, p <0.001). Compared with participants with junior professional titles, the PTS-positive rate of HCWs with intermediate professional titles was 1.91 times greater (90% Cl: 1.35–2.70, p<0.01). Those who had been in contact with confirmed COVID-19 cases were 1.40 times (95% Cl: 1.02–1.92, p <0.05) more likely to have PTS than those who did not have contact with COVID-19 cases or did not know the relevant conditions.

As shown in Fig 3, for depression, the proportion of HCWs with intermediate professional titles was significantly higher, at 1.65 times (90% Cl: 1.17–2.33, p <0.01) that of those with junior professional titles. The depression level of HCWs at work during the epidemic was 1.56 times that of HCWs on vacation (95% Cl: 1.03–2.37, p <0.05), and their anxiety was 1.70 times greater (95% Cl: 1.10–2.63, p <0.05) (Fig 4). Participants who had been in contact with confirmed cases had more pronounced anxiety, 1.40 times that of those who did not have contact with COVID-19 cases or did not know the relevant conditions (95% Cl: 1.02–1.92, p <0.05) (Fig 4). As shown in S3 Fig, the multivariate logistic regression analysis shows that there is no significant correlation between the variables and the positive stress symptoms results.

## Discussion

Among the HCWs participating in the survey, women accounted for 85.02% (1793/2109) and men accounted for 14.98% (316/2109), basically in line with the ratio of males to females in the

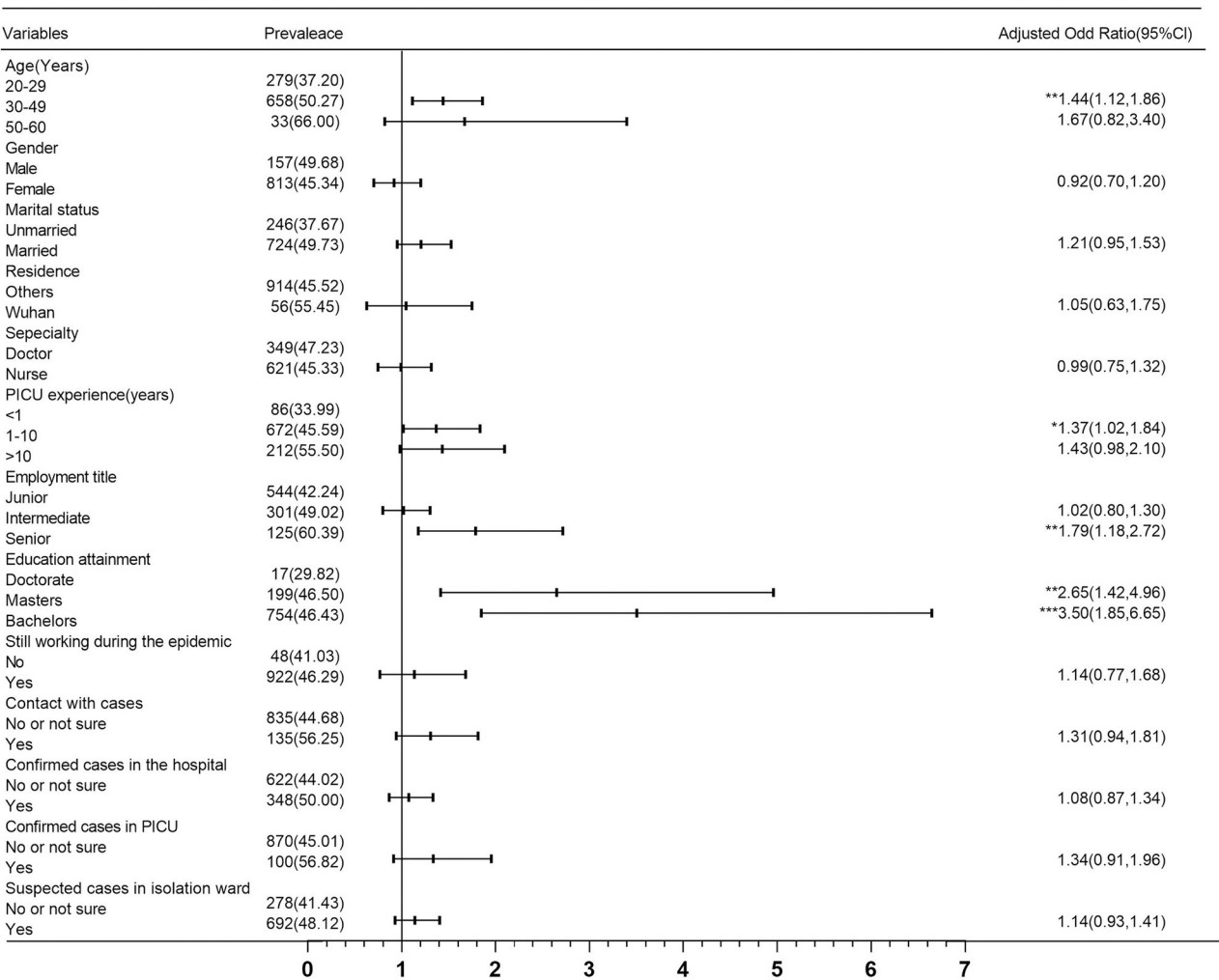

**Fig 2. Multivariable logistic regression models for post-traumatic distress (IES-R≥24) (n = 2109).**

2019 Chinese Health Statistics Yearbook [38]. Therefore, this survey can roughly reflect the psychological distress of PICU HCWs.

In this epidemic, 45.99%, 39.69%, 36.46%, and 17.12% of PICU HCWs had varying degrees of PTS, depression, anxiety, and stress, which were much lower than those of the Brazilian general population in the same study earlier in the epidemic (54.9%, 61.3%, 44.2%, and 50.8%) [30]. At the same time, the report also shows that 84.4% of the research population felt insecure. Given the public's lack of professional knowledge, they were easily confused and driven to fear by a large amount of false information on the Internet; therefore, their psychological status was more vulnerable to the impact of the epidemic. The prevalence of depression, anxiety and stress among participants was higher than that among Chinese HCWs in the same study [33]. To a certain extent [39], this shows that PICU HCWs have a higher degree of psychological influence among all HCWs, and they have more depression, anxiety and stress.

Surprisingly, our research shows that there is no significant difference in the psychological impact among HCWs of different genders, which is inconsistent with many studies [40–47]. In the past, many psychology-related studies have shown that in different groups, not limited

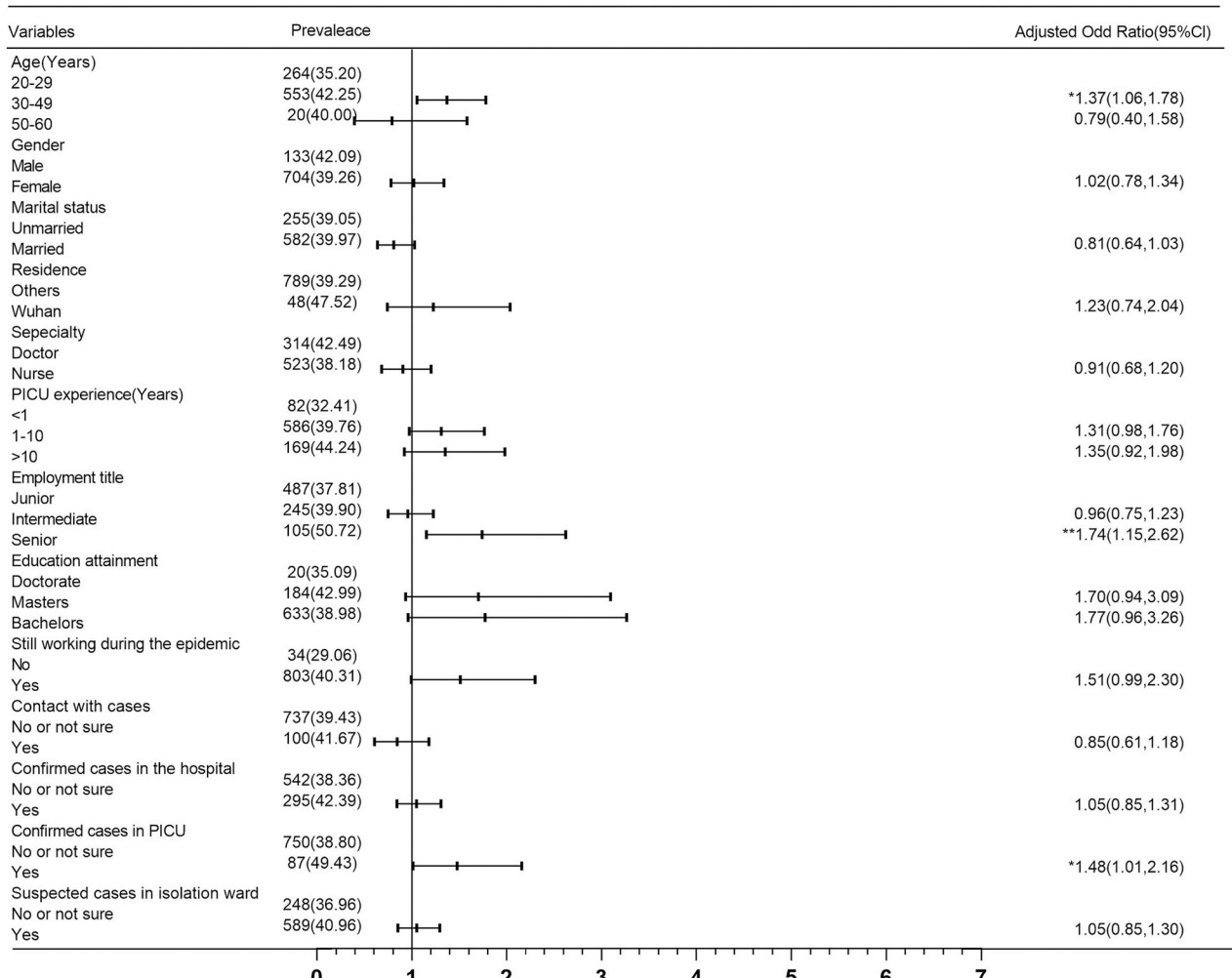

| Variables | Prevaleace | | Adjusted Odd Ratio(95%CI) |
|---|---|---|---|
| Age(Years) | | | |
| 20-29 | 264(35.20) | | |
| 30-49 | 553(42.25) | | *1.37(1.06,1.78) |
| 50-60 | 20(40.00) | | 0.79(0.40,1.58) |
| Gender | | | |
| Male | 133(42.09) | | |
| Female | 704(39.26) | | 1.02(0.78,1.34) |
| Marital status | | | |
| Unmarried | 255(39.05) | | |
| Married | 582(39.97) | | 0.81(0.64,1.03) |
| Residence | | | |
| Others | 789(39.29) | | |
| Wuhan | 48(47.52) | | 1.23(0.74,2.04) |
| Sepecialty | | | |
| Doctor | 314(42.49) | | |
| Nurse | 523(38.18) | | 0.91(0.68,1.20) |
| PICU experience(Years) | | | |
| <1 | 82(32.41) | | |
| 1-10 | 586(39.76) | | 1.31(0.98,1.76) |
| >10 | 169(44.24) | | 1.35(0.92,1.98) |
| Employment title | | | |
| Junior | 487(37.81) | | |
| Intermediate | 245(39.90) | | 0.96(0.75,1.23) |
| Senior | 105(50.72) | | **1.74(1.15,2.62) |
| Education attainment | | | |
| Doctorate | 20(35.09) | | |
| Masters | 184(42.99) | | 1.70(0.94,3.09) |
| Bachelors | 633(38.98) | | 1.77(0.96,3.26) |
| Still working during the epidemic | | | |
| No | 34(29.06) | | |
| Yes | 803(40.31) | | 1.51(0.99,2.30) |
| Contact with cases | | | |
| No or not sure | 737(39.43) | | |
| Yes | 100(41.67) | | 0.85(0.61,1.18) |
| Confirmed cases in the hospital | | | |
| No or not sure | 542(38.36) | | |
| Yes | 295(42.39) | | 1.05(0.85,1.31) |
| Confirmed cases in PICU | | | |
| No or not sure | 750(38.80) | | |
| Yes | 87(49.43) | | *1.48(1.01,2.16) |
| Suspected cases in isolation ward | | | |
| No or not sure | 248(36.96) | | |
| Yes | 589(40.96) | | 1.05(0.85,1.30) |

**Fig 3. Multivariable logistic regression models for depression (DASS-21 depression subscale≥5) (n = 2109).**

to HCWs, women's psychological endurance is weaker than that of men, and their psychological distress is greater [48–50]. This is probably due to our choice of research objects. The participants usually come into contact with patients with life-threatening illnesses, and they are always in a working environment where rescue procedures could be initiated at any time. They are always ready to fight the death, and thus their psychological health may be better than that of those in other departments.

Logistic regression showed that marital status was an independent risk factor for PTS. The COVID-19 epidemic broke out during the Chinese New Year [4]. As a traditional Chinese holiday for family reunions, married HCWs inevitably worried about infecting their families. However, based on the traditional concept of "marrying and giving birth children" in Chinese families, the greater difference between married and unmarried HCWs is the presence of children. As parents, they inevitably have more concerns because at that time, there were very few reports about children with COVID-19, and the diagnosis and treatment of children with COVID-19 had not been unified.

Many studies have shown that professional titles are related to psychological effects [42, 51, 52]. Our research also found that compared with those with junior professional titles, having

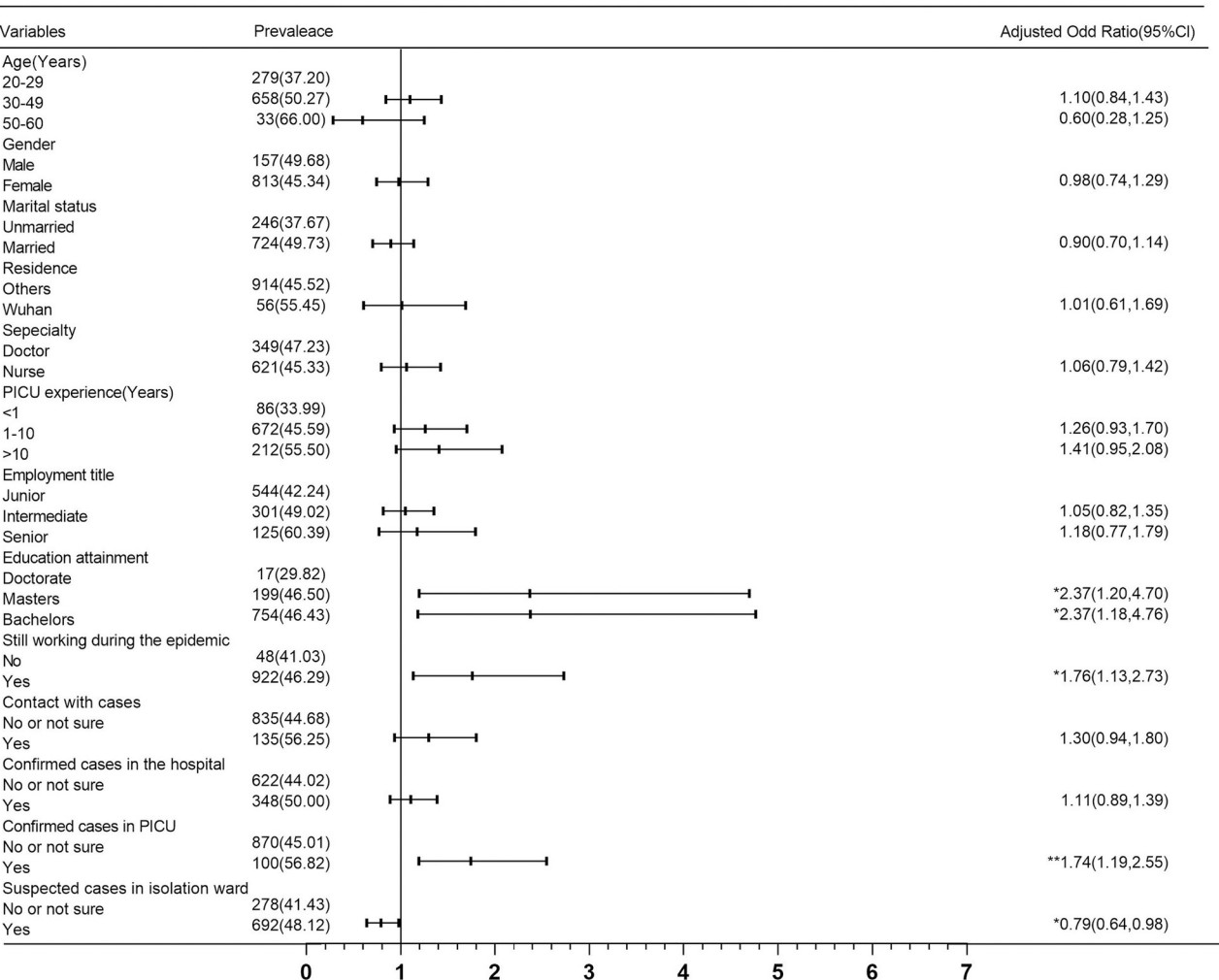

| Variables | Prevaleace | | Adjusted Odd Ratio(95%CI) |
|---|---|---|---|
| Age(Years) | | | |
| 20-29 | 279(37.20) | | |
| 30-49 | 658(50.27) | | 1.10(0.84,1.43) |
| 50-60 | 33(66.00) | | 0.60(0.28,1.25) |
| Gender | | | |
| Male | 157(49.68) | | |
| Female | 813(45.34) | | 0.98(0.74,1.29) |
| Marital status | | | |
| Unmarried | 246(37.67) | | |
| Married | 724(49.73) | | 0.90(0.70,1.14) |
| Residence | | | |
| Others | 914(45.52) | | |
| Wuhan | 56(55.45) | | 1.01(0.61,1.69) |
| Sepecialty | | | |
| Doctor | 349(47.23) | | |
| Nurse | 621(45.33) | | 1.06(0.79,1.42) |
| PICU experience(Years) | | | |
| <1 | 86(33.99) | | |
| 1-10 | 672(45.59) | | 1.26(0.93,1.70) |
| >10 | 212(55.50) | | 1.41(0.95,2.08) |
| Employment title | | | |
| Junior | 544(42.24) | | |
| Intermediate | 301(49.02) | | 1.05(0.82,1.35) |
| Senior | 125(60.39) | | 1.18(0.77,1.79) |
| Education attainment | | | |
| Doctorate | 17(29.82) | | |
| Masters | 199(46.50) | | *2.37(1.20,4.70) |
| Bachelors | 754(46.43) | | *2.37(1.18,4.76) |
| Still working during the epidemic | | | |
| No | 48(41.03) | | |
| Yes | 922(46.29) | | *1.76(1.13,2.73) |
| Contact with cases | | | |
| No or not sure | 835(44.68) | | |
| Yes | 135(56.25) | | 1.30(0.94,1.80) |
| Confirmed cases in the hospital | | | |
| No or not sure | 622(44.02) | | |
| Yes | 348(50.00) | | 1.11(0.89,1.39) |
| Confirmed cases in PICU | | | |
| No or not sure | 870(45.01) | | |
| Yes | 100(56.82) | | **1.74(1.19,2.55) |
| Suspected cases in isolation ward | | | |
| No or not sure | 278(41.43) | | |
| Yes | 692(48.12) | | *0.79(0.64,0.98) |

**Fig 4. Multivariable logistic regression models for anxiety (DASS-21 anxiety subscale≥4) (n = 2109).**

an intermediate professional title was an independent risk factor for PTS and depression. This may be due to an imbalance between the work experience of on one's title and the risk of exposure to cases, the burden and the ability to deal with emotions.

Still working during the epidemic was an independent risk factor for depression and anxiety. On the one hand, HCWs knew very little about the new virus, and they were constantly exploring and learning in the face of cases. This unpredictability greatly increased the workload. On the other hand, HCWs were at a higher risk of exposure to COVID-19 cases. They were more afraid of being infected and infecting others [17, 19, 20]. At the same time, successive reports of HCWs infections struck fear in them. These factors further exacerbated PICU HCW's depression and anxiety.

Exposure history appears controversial as a risk factor [21, 53, 54]. Our research found that exposure to confirmed cases of COVID-19 was a risk factor for PTS and anxiety. This may be due to the different definitions of exposure history in various studies. The exposure history in some studies is defined as exposure to a confirmed or suspected case of COVID-19 [7, 55], which somewhat increased the fear caused by uncertainty and the increase in the positive rate. At the same time, because the contact history differed from the time of the survey, as time

went by, the appearance of corresponding symptoms such as fever and cough also affected the results of the study.

Our research did not find risk factors for stress, but this does not mean that the stress of HCWs during the epidemic was not great. Previous psychological surveys on the different scale about HCWs showed that the stress level of psychological impact during the epidemic was higher than that in normal times [56, 57]. This shows that regardless of whether they were doctors or nurses, their age group and job title, and whether they went to work during the epidemic, HCWs were under more pressure than usual. They remained on the same frontlines to fight the new virus to the end. However, compared with that of the public in other occupations [30, 58], the stress level of HCWs seems to be lower. This is likely due to the economic regression of various industries during the epidemic, which put people in other occupations at greater risk of being laid off.

## Strengths and limitations

First, on March 11, the WHO announced that the COVID-19 outbreak was a pandemic. As of April 1, more than 1 million cases have been confirmed. Our research began on March 26, which was the peak of the global growth rate of COVID-19, and it was basically under control in China. This research occurred during the period when the Chinese epidemic was basically under control, and the global outbreak was officially full-blown; thus, our research has a certain degree of representativeness. Second, China is the country where the first COVID-19 case was discovered. Regarding the unknown and unpredictable nature of the new virus, the challenges faced by Chinese HCWs and the psychological impact they bore merit attention. Finally, this is a large sample multicentre study of all PICU HCWs in China. The sample basically reflects the overall psychological condition of PICU HCWs. However, the study also has certain limitations. On the one hand, it is cross-sectional, and the mental health status of the population is in a continuous process of change. Prospective studies can better determine correlation and causality. On the other hand, the survey site of this study is PICUs in mainland China. The COVID-19 epidemic is a pandemic on a global scale, and there were designated hospitals throughout China during the epidemic; therefore, this study can only represent the psychological status of Chinese PICU HCWs. Finally, because the study was completed voluntarily online, there is a certain level of bias. At the same time, deviation caused by the gender distribution of men and women in the research group cannot excluded.

## Conclusions

In summary, our research shows that during the COVID-19 epidemic, 45.99%, 39.69%, 36.46% and 17.12% of PICU HCWs had varying degrees of PTS, depression, anxiety, and stress, respectively. Exposure history was an independent risk factor for PTS. Having an intermediate professional title and still working during the epidemic were independent risk factors for depression. Still working during the epidemic and COVID-19 contact history were independent risk factors for anxiety. Although the incidence of severe new coronary pneumonia in children is low, the mental health of PICU HCWs should still be considered for early intervention. At the same time, our research provides a certain basis for the occurrence of similar events in the future and early intervention for specific populations.

## Supporting information

**S1 Fig. The number of cases in China as of May 1.**
(TIF)

**S2 Fig. The number of cases in the world as of May 1.**
(TIF)

**S3 Fig. Multivariable logistic regression models for stress (DASS-21 stress subscale≥8) (n = 2109).**
(TIF)

**S1 Table. Percentage of participants with mild to extremely severe PTS, depression, anxiety and stress.**
(DOCX)

**S2 Table. T-tests results for psychological states differences between different age groups, PICU experience and employment title.**
(DOCX)

**S1 Data.**
(XLSX)

## Acknowledgments

The authors thank the participants for their time and effort in filling the questionnaire during these difficult times.

A total of 53 hospitals participated in the questionnaire survey were: Children's Hospital of Chongqing Medical University; Children's Hospital Affiliated to Capital Institute of Pediatrics; Beijing Children's Hospital Affiliated to Capital Medical University; Beijing Fuwai Hospital; Shanghai Children's Medical Center, Shanghai Jiaotong University School of Medicine; Children's Hospital of Fudan University; Xinhua Hospital Affiliated to Shanghai Jiaotong University School of Medicine; Shanghai Children's Hospital; Tianjin Children's Hospital; People's Hospital of Inner Mongolia Autonomous Region; Affiliated Hospital of Inner Mongolia Medical University; Harbin Children's Hospital, Heilongjiang Province; The First Hospital of Jilin University; Shengjing Hospital Affiliated to China Medical University; Dongying People's Hospital, Shandong Province; Qilu Children's Hospital of Shandong University; The First Affiliated Hospital of Shandong First Medical University; Qilu Hospital of Shandong University; Shandong Provincial Hospital Affiliated to Shandong University; The Second Hospital of Hebei Medical University; The Fourth Affiliated Hospital of Hebei Medical University; First Affiliated Hospital of Zhengzhou University; The Third Affiliated Hospital of Zhengzhou University; Zhoukou Central Hospital, Henan Province; Henan Children's Hospital; The First People's Hospital of Datong City, Shanxi Province; Xi'an Children's Hospital; Hanzhong Central Hospital of Shaanxi Province; People's Hospital of Xinjiang Uygur Autonomous Region; People's Hospital of Tibet Autonomous Region; Gansu Province Maternal and Child Health Hospital; General Hospital of Ningxia Medical University; The First Affiliated Hospital of Guangxi Medical University; Guangxi Maternal and Child Health Hospital; Union Hospital, Tongji Medical College, Huazhong University of Science and Technology; Huazhong University of Science Tongji Hospital, Tongji Medical College; Wuhan Children's Hospital; The First Affiliated Hospital of Nanchang University; Hunan Children's Hospital; Hunan Provincial People's Hospital; Children's Hospital of Nanjing Medical University; Children's Hospital of Soochow University; Children's Hospital Affiliated to Zhejiang University School of Medicine; The Second Affiliated Hospital of Wenzhou Medical University, Yuying Children's Hospital; Chengdu Women and Children's Central Hospital; Mianyang Central Hospital; West China Second Hospital of Sichuan University; Women and Children's Hospital of Qinghai Province; The Third Affiliated Hospital of Zunyi Medical University; Maternal and Child Health

Hospital of Guiyang City, Guizhou Province; The First Affiliated Hospital of University of Science and Technology of China; Anhui Children's Hospital; Kunming Children's Hospital; The First People's Hospital of Dali; Xiamen Children's Hospital (Xiamen Branch of Pediatrics Affiliated to Fudan); The First Affiliated Hospital of Xiamen University; Fujian Provincial Maternity and Child Health Hospital; Boai Hospital, Zhongshan City, Guangdong Province; Guangzhou Women and Children's Medical Center; Shenzhen Baoan Maternity and Child Health Hospital Affiliated to Jinan University; Hainan Women and Children's Medical Center; The First Affiliated Hospital of Hainan Medical College.

## Author Contributions

**Conceptualization:** Cheng-Jun Liu, Jing Li, Feng Xu.

**Data curation:** Yue Zhang.

**Formal analysis:** Cheng-Jun Liu, Jing Li, Feng Xu.

**Funding acquisition:** Feng Xu.

**Investigation:** Yue Zhang, Dan-Dan Pi.

**Methodology:** Yue Zhang, Dan-Dan Pi.

**Project administration:** Dan-Dan Pi, Cheng-Jun Liu, Jing Li.

**Resources:** Dan-Dan Pi, Feng Xu.

**Software:** Yue Zhang.

**Supervision:** Dan-Dan Pi, Cheng-Jun Liu, Jing Li, Feng Xu.

**Validation:** Yue Zhang, Cheng-Jun Liu, Jing Li, Feng Xu.

**Visualization:** Yue Zhang.

**Writing – original draft:** Yue Zhang.

**Writing – review & editing:** Dan-Dan Pi, Cheng-Jun Liu, Jing Li, Feng Xu.

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
