## [Decision Letter · Decision Letter 0]

31 Aug 2021

PONE-D-21-13813

Psychological Impact of the 2019 Coronavirus Disease (COVID-19) Epidemic among Medical Workers in China

PLOS ONE

Dear Dr. Xu,

Thank you for submitting your manuscript to PLOS ONE. After careful consideration, we feel that it has merit but does not fully meet PLOS ONE’s publication criteria as it currently stands. Therefore, we invite you to submit a revised version of the manuscript that addresses the points raised during the review process.

An expert in this field has reviewed the manuscript provided constructive comments. I agree with all these comments and encourage the authors revised their work using these good comments. Please remember to prepare a point-by-point response letter. In addition to the reviewer's comments, I would like to authors consider the following relevant references in their revision.

Olashore AA, Akanni OO, Fela-Thomas AL, Khutsafalo K. The psychological impact of COVID-19 on health-care workers in African Countries: A systematic review. Asian J Soc Health Behav 2021;4:85-97

Sharma R, Bansal P, Chhabra M, Bansal C, Arora M. Severe acute respiratory syndrome coronavirus-2-associated perceived stress and anxiety among indian medical students: A cross-sectional study. Asian J Soc Health Behav 2021;4:98-104

We look forward to receiving your revised manuscript.

Kind regards,

Chung-Ying Lin

Academic Editor

PLOS ONE

1. Please ensure that your manuscript meets PLOS ONE's style requirements, including those for file naming. The PLOS ONE style templates can be found at https://journals.plos.org/plosone/s/file?id=wjVg/PLOSOne_formatting_sample_main_body.pdf and https://journals.plos.org/plosone/s/file?id=ba62/PLOSOne_formatting_sample_title_authors_affiliations.pdf.

2. PLOS ONE does not copy edit accepted manuscripts (https://journals.plos.org/plosone/s/criteria-for-publication#loc-5). To that effect, please ensure that your submission is free of typos and grammatical errors.

Additional Editor Comments (if provided):

Reviewers' comments:

Reviewer's Responses to Questions

**Comments to the Author**

1. Is the manuscript technically sound, and do the data support the conclusions?

Reviewer #1: Partly

2. Has the statistical analysis been performed appropriately and rigorously? 

Reviewer #1: Yes

3. Have the authors made all data underlying the findings in their manuscript fully available?

Reviewer #1: Yes

4. Is the manuscript presented in an intelligible fashion and written in standard English?

Reviewer #1: No

5. Review Comments to the Author

Reviewer #1: Reviewer#

Comments to the Author

Summary:

The study aimed to explore the psychological impact of the COVID-19 outbreak on

Medical workers and analyses their independent risk factors. The data were based on 2123 medical workers, and the findings show that during the COVID-19 epidemic, the mental health status of

PICU medical workers were affected to varying degrees in terms of stress, depression

and anxiety. Marital status, job title, education level, working status and degree of

contact with COVID-19 were independent risk factors.

However, there still were some notable deficiencies in the article.

Abstract

1. When writing about PICUs for the first time, the author should give its full name or meaning?

Introduction

2.Page 4, lines 71-72. Please briefly revisit and summarize the research gaps. There is no link between the study aim and the previous sentence statement.

Method:

3. Page 4, Line 87-90 The reason for not taking consent is unclear from the sentence.

4. Are there any inclusion and exclusion criteria for sample size?

5. What is the value of Cronbach's α? How does the author has collected the data of Cronbach's α values in the present study?

Results:

6. Page 6, Line 127 to 136 It is unclear what percentage and number mentioned are for what? The author needs to rephrase the sentence.

7. Please provide legends in the table for better understanding.

Discussion

8. There is no real discussion with the data; it is very descriptive. The discussion should not simply discuss the similarities and differences with previous studies.

9. Author need to provided citations whenever they are comparing their finding with prior research. E.g. page 15, line 199 to 202 and 207 to 213

10. Page 16, Line 227 to 229 what are these scores, and how is it related to social responsibilities?

11. It is hard to understand the discussion point made by the author. I think the author needs to make it understandable to readers with proper citation wherever required.

12. Author has just mentioned about limitation of the study; please provide strength

13. Author has mentioned limitations in conclusion. Authors are suggested to phrase a subheading name "Strengths and limitations" after discussion, and within this subsection, put the strengths first, followed by the study's limitations.

14. I see the English writing of this paper can be improved, not only grammar but the way how the statements were phrased were not concise enough.

6. PLOS authors have the option to publish the peer review history of their article (what does this mean?). If published, this will include your full peer review and any attached files.

Reviewer #1: No

---

## [Author Response · Author response to Decision Letter 0]

15 Oct 2021

Dear Editor,

On behalf of my-authors, we thank you very much for giving us an opportunity to revise our manuscript. We appreciate editor and reviewers very much for their positive and constructive comments and suggestions on our manuscript entitled “Psychological Impact of the COVID-19 Epidemic among Healthcare Workers in paediatric intensive care units in China” (manuscript PONE-D-21-13813). To address the critiques of the reviewers, we revised our manuscript according to their comments. Attached please find the revised version (All changes are marked as red color), which we would like to submit for you kind consideration. We would like to express our great appreciation to you and reviewers for comments on our paper.

Looking forward to hearing from you. Thank you and best regards.

Yours sincerely,

Correspondence to:

Yue Zhang

18375763857@163.com

Abstract

1. When writing about PICUs for the first time, the author should give its full name or meaning?

A: We have given its full name.

Introduction

2.Page 4, lines 71-72. Please briefly revisit and summarize the research gaps. There is no link between the study aim and the previous sentence statement.

A: At present, research on the psychological effects of PICU HCWs is very limited. We have re-organized and explained the reasons for choosing PICU HCWs as the research object.

Method:

3. Page 4, Line 87-90 The reason for not taking consent is unclear from the sentence.

A: All participants voluntarily responded to the survey anonymously and provided informed consent online before the survey.

4. Are there any inclusion and exclusion criteria for sample size?

A: Since the questionnaire was completed voluntarily, there are no inclusion and exclusion criteria.

5. What is the value of Cronbach's α? How does the author has collected the data of Cronbach's α values in the present study?

A: The scale we chose is a mature Liszt scale, so there is no calculation about reliability and validity.

Results:

6. Page 6, Line 127 to 136 It is unclear what percentage and number mentioned are for what? The author needs to rephrase the sentence.

A: We have rephrased the sentence.

7. Please provide legends in the table for better understanding.

A: We have provided legends in the table.

Discussion

8. There is no real discussion with the data; it is very descriptive. The discussion should not simply discuss the similarities and differences with previous studies.

9. Author need to provided citations whenever they are comparing their finding with prior research. E.g. page 15, line 199 to 202 and 207 to 213

10. Page 16, Line 227 to 229 what are these scores, and how is it related to social responsibilities?

11. It is hard to understand the discussion point made by the author. I think the author needs to make it understandable to readers with proper citation wherever required.

A: On the basis of past research, we have re-discussed and analyzed the data.

12. Author has just mentioned about limitation of the study; please provide strength

13. Author has mentioned limitations in conclusion. Authors are suggested to phrase a subheading name "Strengths and limitations" after discussion, and within this subsection, put the strengths first, followed by the study's limitations.

A: We have phrased a subheading name "Strengths and limitations" after discussion.

14. I see the English writing of this paper can be improved, not only grammar but the way how the statements were phrased were not concise enough.

A: We have submitted our manuscript to “AJE” for language polishing.

---

## [Decision Letter · Decision Letter 1]

16 Nov 2021

PONE-D-21-13813R1Psychological Impact of the COVID-19 Epidemic among Healthcare Workers in paediatric intensive care units in ChinaPLOS ONE

Dear Dr. Xu,

Thank you for submitting your manuscript to PLOS ONE. After careful consideration, we feel that it has merit but does not fully meet PLOS ONE’s publication criteria as it currently stands. Therefore, we invite you to submit a revised version of the manuscript that addresses the points raised during the review process.

The reviewer found that all the prior concerns have been addressed. However, some minor revisions are needed. Please revise your manuscript accordingly. Thank you.==============================

We look forward to receiving your revised manuscript.

Kind regards,

Chung-Ying Lin

Academic Editor

PLOS ONE

Journal Requirements:

Reviewers' comments:

Reviewer's Responses to Questions

**Comments to the Author**

1. If the authors have adequately addressed your comments raised in a previous round of review and you feel that this manuscript is now acceptable for publication, you may indicate that here to bypass the “Comments to the Author” section, enter your conflict of interest statement in the “Confidential to Editor” section, and submit your "Accept" recommendation.

Reviewer #1: All comments have been addressed

2. Is the manuscript technically sound, and do the data support the conclusions?

Reviewer #1: Yes

3. Has the statistical analysis been performed appropriately and rigorously? 

Reviewer #1: Yes

4. Have the authors made all data underlying the findings in their manuscript fully available?

Reviewer #1: Yes

5. Is the manuscript presented in an intelligible fashion and written in standard English?

Reviewer #1: No

6. Review Comments to the Author

Reviewer #1: In general, my previous comments have been addressed and implemented appropriately in the manuscript. There are a few minor comments that I hope that the authors could address:

1. Please carefully check the citation required in the manuscript for eg. Line no. 424-427 page 20 and Line 434- 434

2. What does author mean by the positive rate of HCWs seems to be lower in Line no. 438 page 20?

3. Line no. page 20 What does author mean by following sentence. “However, the greater difference between married and unmarried HCWs is children.” Kindly rephrase it to make it meaningful.

4. Further, I would like to suggest author to change the first paragraph of discussion, its similar to the introduction.

7. PLOS authors have the option to publish the peer review history of their article (what does this mean?). If published, this will include your full peer review and any attached files.

Reviewer #1: No

---

## [Author Response · Author response to Decision Letter 1]

15 Feb 2022

Dear Reviewer,

On behalf of my-authors, we thank you very much for giving us an opportunity to revise our manuscript. We appreciate you very much for positive and constructive comments and suggestions on our manuscript entitled “Psychological Impact of the COVID-19 Epidemic among Healthcare Workers in paediatric intensive care units in China” (manuscript PONE-D-21-13813). To address the critiques, we revised our manuscript according to comments. We have submitted our manuscript to “AJE” for premium editing again and will respond to each point. Attached please find the revised version (All changes are marked as red color), which we would like to submit for you kind consideration. We would like to express our great appreciation to you for comments on our paper.

Looking forward to hearing from you. Thank you and best regards.

Yours sincerely,

Correspondence to:

Yue Zhang

18375763857@163.com

---

## [Editor Report · Decision Letter 2]

2 Mar 2022

Psychological Impact of the COVID-19 Epidemic among Healthcare Workers in paediatric intensive care units in China

PONE-D-21-13813R2

Dear Dr. Xu,

We’re pleased to inform you that your manuscript has been judged scientifically suitable for publication and will be formally accepted for publication once it meets all outstanding technical requirements.

Kind regards,

Chung-Ying Lin

Academic Editor

PLOS ONE
---

## [Editor Report · Acceptance letter]

11 Mar 2022

PONE-D-21-13813R2 

Psychological Impact of the COVID-19 Epidemic among Healthcare Workers in paediatric intensive care units in China 

Dear Dr. Xu:

I'm pleased to inform you that your manuscript has been deemed suitable for publication in PLOS ONE. Congratulations! Your manuscript is now with our production department. 

Kind regards, 

on behalf of

Dr. Chung-Ying Lin 

Academic Editor

PLOS ONE